# Strain Modified Constitutive Equation and Processing Maps of High Quality 20MnCr5(SH) Gear Steel

**Jingcheng Yang [1], Lizhong Wang [1,2,\*], Yingjun Zheng [3] and Zhiping Zhong [3]**

1    School of Mechanical Engineering, Xinjiang University, Urumqi 830047, China; yangjingcheng0705@163.com
2    State Key Laboratory of Mechanical Manufacturing System Engineering, Xi'an Jiaotong University, Xi'an 710049, China
3    Tai Cang Jiuxin Precision Toolings Co., Ltd., Suzhou 215400, China; jj@jiujin.org (Y.Z.); zhongzhiping@jiujin.org (Z.Z.)
*    Correspondence: wanglz@mail.xjtu.edu.cn; Tel.: +86-137-7203-4988

**Abstract:** In order to develop the high-temperature forging process of high-quality 20MnCr5(SH) gear steel, according to the physical characteristics of high-temperature hot deformation of 20MnCr5(SH), the single pass hot pressing test was carried out in the temperature range of 930–123 °C and the strain rate range of 0.002–2 s$^{-1}$ by using a Gleeble-1500D thermal simulator. The stress-strain curve of 20MnCr5(SH) was analyzed and confirmed by microstructure analysis. The dynamic recrystallization occurred, and the constitutive equation of 20MnCr5(SH) high temperature flow stress was established. Considering that the traditional Arrhenius constitutive equation does not consider the effect of strain on the constitutive equation, a strain modified Arrhenius constitutive equation is proposed. The results show that the correlation is 0.9895 and the average relative error is 8.048%, which verifies the stress prediction ability of the strain modified constitutive equation. According to the dynamic material theory and instability criterion, the processing maps of 20MnCr5(SH) are obtained. It is therefore considered that 20MnCr5(SH) is most suitable for thermoplastic processing at strain rate of 0.05–1 s$^{-1}$ and temperature of 1030–1100 °C.

**Keywords:** 20MnCr5(SH) gear steel; hot compression test; constitutive equation; strain compensation; microstructure; processing maps

## 1. Introduction

20MnCr5 steel is German standard alloy structural steel, which is equivalent to Chinese standard 20CrMn steel. 20MnCr5 increases the sulfur content, which makes the cutting performance better, while leaving the welding performance poor. 20MnCr5 is generally used after carburizing, quenching and tempering, and the properties after heat treatment are better than 20Cr [1]. 20MnCr5 can be used to manufacture quenched and tempered parts with large cross-section and carburized parts with small cross-section. It can also be used to manufacture medium and small parts with medium load and small impact, and can replace 20CrNi. For example, gears, shafts, spindles, friction wheels of variable speed equipment, worms, sleeves of speed governors, etc. 20MnCr5 is widely used in automobile gears and occupies a large market share [2]. By optimizing slag proportioning structure, the quality of 20MnCr5 steel was further improved, and a high quality 20MnCr5(SH) gear steel was produced. This kind of gear steel has good hardenability. The hardenability width of high quality 20MnCr5(SH) gear steel after optimized smelting process reaches △HRC ≤3. It has high forging qualification rate, strong heat treatment stability, small carburizing and quenching deformation, and excellent processing performance [3].

The constitutive equation of metal material is a mathematical model reflecting the inherent properties of metal and revealing the mechanism of plastic deformation [4]. Johnson-cook model (J-C model) is the most widely used cold plastic constitutive equation of metals at room temperature [5]. At high temperature, Zerilli–Armstrong model (Z-A

model) and Arrhenius hyperbolic sine model are applied. The Z-A model is not as easy to use as the Arrhenius model because it needs to know the structure of the metal unit cell to select the appropriate parameters [6].

The traditional Arrhenius hyperbolic sine constitutive equation mainly considers the influence of peak stress on the constants of the constitutive equation, which is far from enough [7]. In the process of establishing the constitutive equation, the stress corresponding to different strain variables should be taken into account. The strain modification of the Arrhenius constitutive equation is theoretically beneficial to improve the accuracy of flow stress prediction. Wang et al. [8] established the high temperature flow stress constitutive equation for alloy transition layer in composite steel pipe, and expressed the material parameters in Arrhenius constitutive equation with quintic equation group, and the fitting correlation is above 0.97847. Zhao et al. [9] carried out a hot compression experiment on 40CrNi steel, and established octave equations about strain according to material parameters. Finally, the average relative error of modified prediction of the strain-modified Arrhenius constitutive equation was only 3.4%; Lei et al. [10] established a strain compensation constitutive equation of 3Cr-1Si-1Ni ultra high strength steel. The reciprocal of a material constant in. The correlation between the predicted values and the experimental values was as high as 0.994.

The processing map is a very effective tool to study the thermoplastic processing properties of metal materials [11]. The processing map can judge the processing efficiency in a certain process window from the perspective of energy direction. According to the dynamic material model, Jia at al. [12] established the processing maps of 2219 aluminum alloy, finding that the peak power dissipation rate reached 0.336, and one of the peak power dissipation rate regions was in a region with high temperature and high strain rate, indicating that the region suitable for thermoplastic processing was not necessarily the region with high temperature and low strain rate. Cai et al. [13] Studied the hot deformation behavior of high nitrogen martensitic stainless steel 0.3C-15Cr-1Mo-0.5N, and also studied the hot plastic processing properties of 0.3C-15Cr-1Mo-0.5N from the perspectives of dynamic recrystallization kinetics and processing map. The most suitable hot processing regions are at 1303 K–1423 K and strain rate 5 s$^{-1}$–10 s$^{-1}$, respectively.

At present, the research on 20MnCr5 gear steel reported in the literature mainly involves fatigue failure behavior [14], viscoplastic properties [15] and surface treatment [16] of 20MnCr5 gear steel. The hot workability of 20MnCr5 gear steel has also been studied from the perspective of dynamic recrystallization kinetics, but it is not comprehensive to evaluate the hot workability only by dynamic recrystallization behavior [17]. The most intuitive method to reflect the hot workability of metal materials is the processing map. At the same time, the microscopic mechanism constitutive model based on the dynamic recrystallization kinetic model is not as good as Arrhenius phenomenological constitutive model, which is convenient for the secondary development of finite element simulation software [18].

In order to better study the hot plastic forming process of high quality 20MnCr5(SH) gear steel, it is necessary to establish Arrhenius constitutive equation for the secondary development of relevant finite element simulation software; In order to make up for the theoretical defect that the Arrhenius constitutive equation only considers peak stress and does not consider strain, it is necessary to establish a modified constitutive equation for strain compensation; In order to comprehensively and intuitively evaluate high quality. It is necessary to establish the processing map of high quality 20MnCr5(SH) gear steel. In this paper, the hot deformation behavior and hot workability of high quality 20MnCr5(SH) gear steel will be systematically studied.

## 2. Materials and Experimental Procedure

The material used in the test is 20MnCr5(SH) round bar without any heat treatment, and the chemical composition of the material is shown in Table 1. The sample is turned

into a cylinder with a size of 8 mm × 12 mm, and the end face is polished to a roughness of Ra1.6 by a grinder to prevent excessive test error caused by end face defects.

**Table 1.** Contents of various chemical elements in 20MnCr5(SH) gear steel (%, mass fraction).

| C | Si | Mn | P | S | Cr | Si | Al | Cu |
|---|---|---|---|---|---|---|---|---|
| 0.17–0.22 | ≤0.12 | 1.00–1.50 | ≤0.035 | 0.01–0.035 | 0.80–1.30 | ≤0.12 | 0.02–0.04 | ≤0.20 |

The experiment was carried out on Gleeble-1500D thermo-mechanical simulator (Data Sciences International, St. Paul, MN, USA). The heating rate of the specimen was set at 10 °C/s. After the sample was heated to the experimental temperature, it was kept for about 3 min to make the internal temperature distribution of the sample uniform. Then the hot compression experiment was carried out. In this hot compression experiment, the temperature was 4 groups, the gradient was 100 °C, the strain rate was 0.002, 0.02, 0.2, 2 s$^{-1}$, and the strain was 0.916. After the test, the hot compression deformation structure was retained by water quenching. The temperature change during the test is shown in Figure 1.

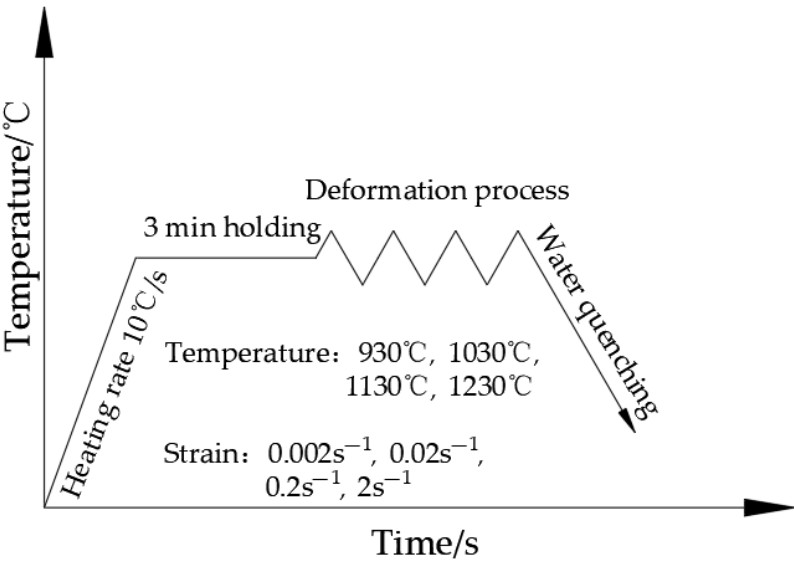

**Figure 1.** Temperature curve of hot compression test.

The cooled specimens were cut along the axial direction by metallographic cutting machine, polished by grinding machine and polishing machine, the equipment used for sample grinding and polishing is Struers Tegrapol-21 (Zeiss, Jena, Germany). Finally, they were corroded by Saturated picric acid aqueous solution, configuration method: dissolve 1.5 g picric acid solid in 100 mL distilled water. The microstructure of 20MnCr5(SH) was observed by metallographic microscope Nikon MA100 (Zeiss, Jena, Germany).

### 3. Results and Analysis

*3.1. Flow Stress Curve Analysis*

The true stress-strain curves of 20MnCr5(SH) gear steel at different temperatures and strain rates are shown in Figure 2. The real stress-strain curves in Figure 2 show typical dynamic recrystallization. When the strain is small, the flow stress increases rapidly due to the work hardening effect. With the increase of softening effect caused by dynamic recrystallization, the growth rate of flow stress slows down until the peak value of the stress-strain curve appears. At this time, the softening behavior caused by recrystallization began to dominate, and the flow stress of 20MnCr5(SH) gear steel began to decrease. From the four curves of 1030 °C, 1130 °C and 1230 °C in Figure 2a,c, it can be observed that the

stress rises and falls again after the decrease of stress, but the stress peak formed by the increase and fall of stress again is not as large as the previous stress peak, which may be due to the secondary hardening of 20MnCr5(SH) after the end of recrystallization, and recrystallization occurs again after the secondary hardening. Although the other curves did not produce an obvious second stress peak, they all appeared secondary hardening phenomenon, which may be because the temperature and strain rate are not enough to support the recrystallization softening to form the stress peak after secondary hardening in the range of strain 0–0.916.

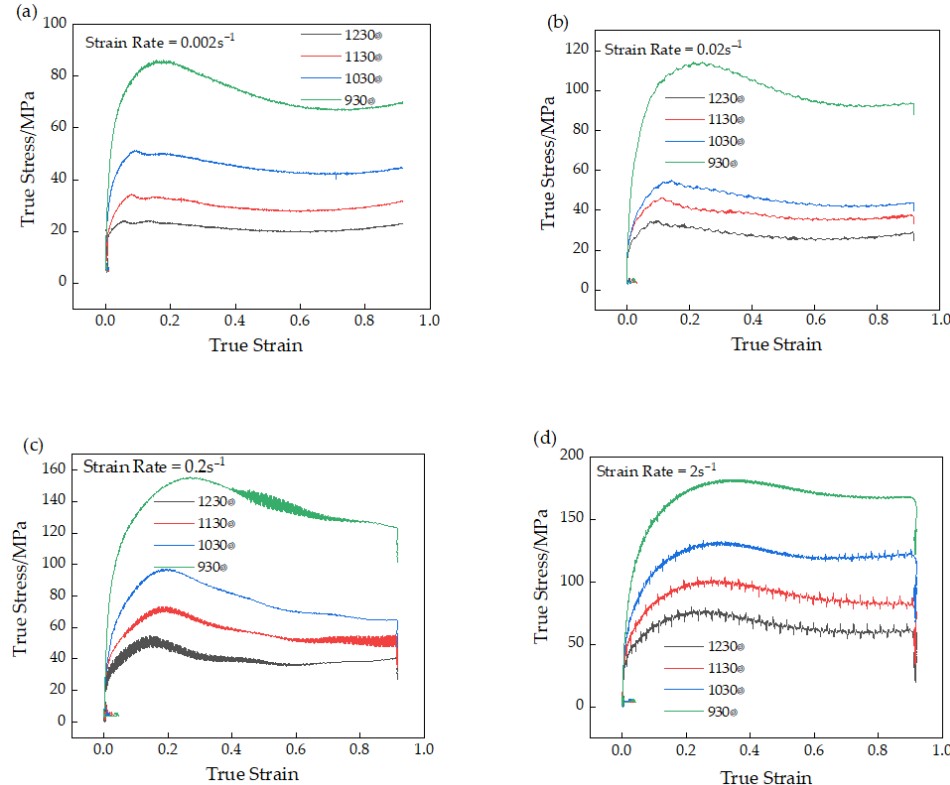

**Figure 2.** Stress-strain curves of 20MnCr5(SH) gear steel at different strains with temperature of (**a**) 0.002 s$^{-1}$; (**b**) 0.02 s$^{-1}$; (**c**) 0.2 s$^{-1}$; (**d**) 2 s$^{-1}$.

The stress curve in Figure 2 is wave-shaped, which may be caused by discontinuous dynamic recrystallization, and the softening effect and processing hardening effect caused by dynamic recovery and dynamic recrystallization are dominant.

The strain values corresponding to the peak stress of 20MnCr5 (SH) gear steel are shown in Table 2. It can be observed that when the temperature is the same and the strain rate is different, the lower the strain rate is, the smaller the peak strain. This may be due to the large strain rate, which leads to the formation of a large number of new dislocations in the grains of the metal, which then makes the work hardening effect significant and delays the arrival of the peak stress.

**Table 2.** Strain value corresponding to peak stress.

| $\dot{\varepsilon}/s$ | 930 °C | 1030 °C | 1130 °C | 1230 °C |
|---|---|---|---|---|
| 0.002 | 0.17765 | 0.09375 | 0.07893 | 0.05331 |
| 0.02 | 0.24110 | 0.14195 | 0.11200 | 0.10092 |
| 0.2 | 0.26796 | 0.19074 | 0.19030 | 0.14090 |
| 2 | 0.35066 | 0.25377 | 0.27823 | 0.21607 |

The variation of peak stress with temperature is shown in Figure 3. It can be found that the higher the temperature is, the smaller the deformation resistance, the smaller the strain rate, and the smaller the deformation resistance.

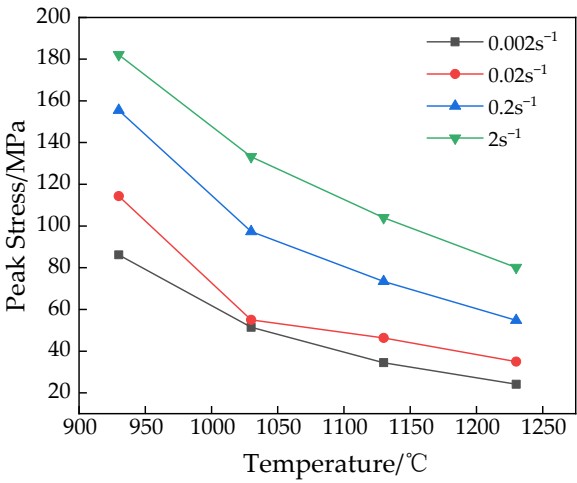

**Figure 3.** Variation of peak stress with temperature corresponding to different strain rates.

### 3.2. Microstructure Observation

The microstructure of some specimens of 20MnCr5(SH) gear steel during hot deformation is shown in Figure 4. The corresponding strain of metallographic diagram is 0.916, and the corrosion reagent used is saturated picric acid solution. It can be seen from Figure 4a–e that the saturated picric acid solution can make the austenite grain boundary of 20MnCr5 (SH) gear steel appear during hot deformation. The grain boundary in Figure 4. is distorted because of the severe deformation of the grain boundary caused by the hot compression process. In Figure 4a–e, due to excessive corrosion in some grains, martensite lath bundles are shown in the grains. It can be seen that the microstructure of 20MnCr5(SH) gear steel formed by hot deformation is lath martensite. Figure 4a–c shows the phase diagram of 20MnCr5(SH) alloy with different strain rates at 1230 °C. It can be found that the metallographic diagram with a strain rate of 0.2 s$^{-1}$ represented in Figure 4b clearly shows that some smaller grains are distributed around the larger grains, which indicates the dynamic recrystallization behavior of 20MnCr5(SH) during hot compression. Compared with the grain size in Figure 4b, the grain size in Figure 4a with a strain rate of 0.02 s$^{-1}$ is larger, which indicates that the grains produced by recrystallization are fully grown, and it is difficult to distinguish the original grains from the recrystallized grains. The grain size in the strain rate of 0.002 s$^{-1}$ represented in Figure 4c is larger than that in Figure 4a, and the recrystallized grains grew more fully. The above phenomenon shows that the dynamic recrystallization behavior of grains can be more fully carried out with the decrease of strain rate. Figure 4c–e are the phase diagram of 20MnCr5(SH) with a strain rate of 0.002 s$^{-1}$ but different temperatures. The recrystallized grains that have not grown up can be seen in Figure 4e. The recrystallized grains in Figure 4d and the original grains have grown up, but there is a certain gap between the recrystallized grains and the original grains, which can be distinguished. In Figure 4c, the grains grow further, and it is difficult to distinguish the recrystallized grains from the original grains. For metal materials, the thermoplastic processing performance of metal is the best when recrystallization occurs and the grain size is small. It is easy to find that the sample with the highest temperature and the lowest strain rate in Figure 4c has larger grain size, which is not the ideal microstructure for thermoplastic processing. The evaluation of hot workability needs further analysis.

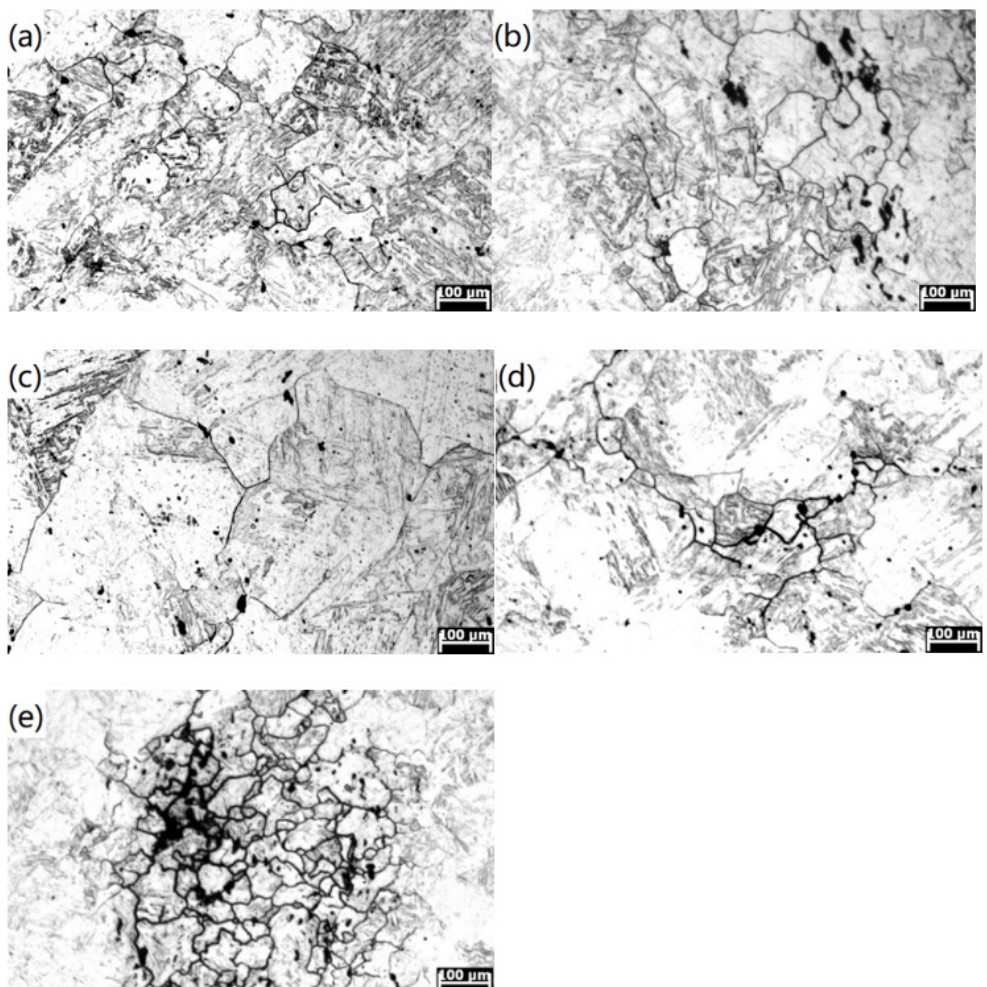

**Figure 4.** The microstructures of the specimens deformed at: (**a**) 1230 °C, $\dot{\varepsilon}$ = 0.02 s$^{-1}$; (**b**) 1230 °C, $\dot{\varepsilon}$ = 0.2 s$^{-1}$; (**c**) 1230 °C, $\dot{\varepsilon}$ = 0.002 s$^{-1}$; (**d**) 1130 °C, $\dot{\varepsilon}$ = 0.002 s$^{-1}$; (**e**) 1030 °C, $\dot{\varepsilon}$ = 0.002 s$^{-1}$.

## 4. Establishment and Verification of Modified Constitutive Equation

Sellars and Tegart [19,20] and other experts and scholars have studied, analyzed and established Arrhenius hyperbolic sine constitutive equation which can scientifically and effectively describe the thermal deformation behavior of metal materials. It is considered that the flow stress, temperature and strain rate can be characterized by Equation (1) including the determined activation energy $Q_{act}$ and temperature $T$:

$$\dot{\varepsilon} = A F(\sigma) \exp\left(\frac{Q_{act}}{RT}\right) \tag{1}$$

where $Q_{act}$—deformation activation energy (kJ/mol); F($\sigma$)—function of stress $\sigma$; R—gas constant (8.314 J/mol·K); $T$—temperature/K; $A$—material constant; $\dot{\varepsilon}$—strain rate/s$^{-1}$ [21,22].

F ($\sigma$) is a function of stress $\sigma$, which exists as follows [19,20]:

$$F(\sigma) = \sigma^{n_1} \quad (\alpha\sigma < 0.8) \tag{2}$$

$$F(\sigma) = \exp(\beta\sigma) \ (\alpha\sigma > 1.2) \tag{3}$$

$$F(\sigma) = [\sin h(\alpha\sigma)]^n. \ \text{(for all stress)} \tag{4}$$

where $\alpha = \beta/n_1$; $\sigma$—flow stress/MPa; $\beta$—material constant; $n$, $n_1$—strain hardening index.

According to the research conclusions of Zener and Hollomon, when metal materials undergo plastic deformation at high temperature, the activation process of thermal defor-

mation controls the strain rate, and the relationship between strain rate and temperature can be expressed by $Z$ parameter [23]:

$$Z = \dot{\varepsilon}\exp(\frac{Q_{act}}{RT}) = A[\sin h(\alpha\sigma_p)]^n \tag{5}$$

where $\sigma_p$—peak stress/MPa; $Z$—temperature and strain rate compensation factor ($Z$ parameter)

By substituting Equations (2)–(4) into Equation (1), and taking the natural logarithm of the left and right sides of the three equations at the same time, we can get the following results [24]:

$$\ln\dot{\varepsilon} = \ln A_1 + n_1\ln\sigma - \frac{Q_{act}}{RT} \tag{6}$$

$$\ln\dot{\varepsilon} = \ln A_2 + \beta\sigma - \frac{Q_{act}}{RT} \tag{7}$$

$$\ln\dot{\varepsilon} = \ln A + n\ln[\sin h(\alpha\sigma)] - \frac{Q_{act}}{RT} \tag{8}$$

$\ln\dot{\varepsilon}$—$\ln\sigma_p$ relation curve and $\ln\dot{\varepsilon}$—$\sigma_p$ relation curve are obtained by linear fitting method for Equations (6)–(8), respectively, as shown in Figure 5a,b. According to the mean value of linear fitting slope of $\ln\dot{\varepsilon}$—$\ln\sigma$ relation curve and $\ln\dot{\varepsilon}$—$\sigma_p$ relation curve, $n_1 = 6.7185125$, $\beta = 0.0891825$ MPa$^{-1}$, $\alpha = \beta/n_1 = 0.013274123$ can be obtained.

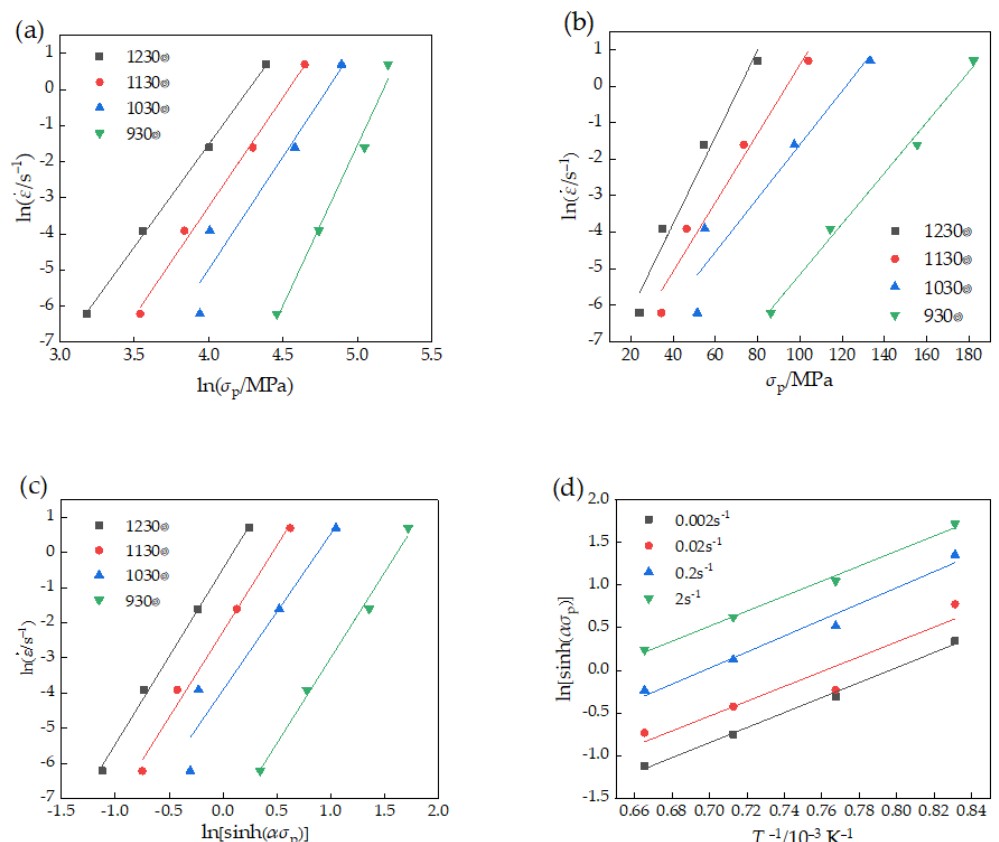

**Figure 5.** The linear relationships of (**a**) $\ln\dot{\varepsilon}$–$\ln\sigma_p$; (**b**) $\ln\dot{\varepsilon}$–$\sigma_p$; (**c**) $\ln\dot{\varepsilon}$–$\ln[\sinh(\alpha\sigma_p)]$; (**d**) $1000/T$–$\ln[\sinh(\alpha\sigma_p)]$.

The strain hardening exponent $n = 4.802$ can be obtained from the relation curve of $\ln\dot{\varepsilon}$—$\ln\sinh(\alpha\sigma_p)$ in Figure 5c. The calculation equation of the deformation activation energy is shown in Equation (9):

$$Q_{act} = R \times \left| \frac{\partial \ln\dot{\varepsilon}}{\partial \ln[\sin h(\alpha\sigma_p)]} \right| \times \left| \frac{\partial \ln[\sin h(\alpha\sigma_p)]}{\partial (1000/T)} \right| \tag{9}$$

Combined with the mean value of linear fitting slope in Figure 5c,d and substituted into Equation (9), the thermal deformation activation energy $Q_{act}$ of 20MnCr5(SH) can be calculated as 356.41997kJ/mol. Then take logarithm on both sides of Equation (5) to get the following equation:

$$\ln\dot{\varepsilon} + \frac{Q_{act}}{RT} = \ln A + n\ln[\sin h(\alpha\sigma_p)] \tag{10}$$

At this time, it can be found that Equation (10) reflects the linear relationship of Figure 5c, and the slope is $n = 4.802$, so the intercept of Equation (10) is $\ln A - Q_{act}/RT$, and the mean intercept has been obtained from Figure 5c, so it can be solved that $\ln A = 22.21909$, $A = 1.85662 \times 10^{12}$.

The relationship between $\ln Z$ and $\ln\sinh(\alpha\sigma_p)$ is shown in Figure 6. The Arrhenius flow stress constitutive equation of 20MnCr5(SH) during hot deformation can be obtained by substituting various material constants into Equation (8), which is shown in Equation (11):

$$\dot{\varepsilon} = 1.857 \times 10^{12} \left[\sin h(0.013\sigma_p)\right]^{4.802} \exp\left(\frac{-356420}{8.314T}\right) \tag{11}$$

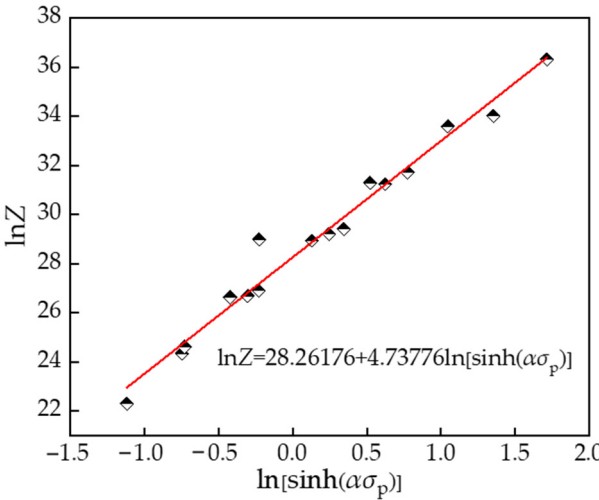

**Figure 6.** The linear relationships of $\ln[\sinh(\alpha\sigma_p)]$ and $\ln Z$.

The traditional Arrhenius flow stress constitutive equation simply uses peak stress to describe the relationship between stress and strain at different temperatures and strain rates. The material parameters in the default model are all constant, and the effect of strain on the constitutive model is not considered in the process of modeling. Various evidences show that each material constant in the constitutive equation is not a fixed value, but a mapping relationship with strain [25,26]. Therefore, it is necessary to modify the traditional Arrhenius constitutive equation.

In order to better reflect the prediction ability of the strain modified constitutive equation, this paper divides the data used to construct the strain modified constitutive equation into two groups, one used to construct the strain modified constitutive equation, and the other to test the accuracy of the strain modified constitutive equation. Because the data used to test the strain correction constitutive equation is not used to build the strain correction constitutive equation, using this method to test the strain correction constitutive

equation can more objectively reflect the prediction ability of the constitutive equation. In this paper, the data of 1130 °C is selected as the test group, and the rest of the data is used to build the strain correction constitutive equation. The data of the test group accounts for 25% of the total data.

According to the calculation method of traditional Arrhenius constitutive equation in the previous section, in the strain range of 0.1–0.9, the stress values corresponding to 9 groups of strains are read from the hot compression experimental data at 0.1 intervals. The polynomial fitting relationship between the strain and four material constants ($\alpha$, $n$, $Q_{act}$, $\ln A$) is established by fitting the curves of four material constants with strain. Because polynomial order plays a decisive role in fitting accuracy, it is particularly important to study the influence of polynomial order on prediction accuracy of strain correction constitutive equation. In this paper, we will study the fitting conditions of order 3, 4, 5 and 6, and take the stress prediction accuracy as the evaluation index to study the influence of polynomial order on stress prediction. The order of polynomial is too low, for example, the order of 1, 2 can't reflect the trend of four material constants with strain; If the order of polynomial is higher than 6, overfitting may be caused.

In order to evaluate the prediction accuracy of strain compensation constitutive equation for flow stress, the correlation coefficient $R$ and relative error AARE are introduced as the evaluation indexes to measure the accuracy of strain correction constitutive model. The calculation equations of the two parameters are (12) and (13), respectively. Where $E_i$ is the experimental value, $P_i$ is the calculated value, $\overline{E_i}$ and $\overline{P_i}$ are the average of the experimental value and the calculated value, respectively. The data in Table 3 are the average values of correlation coefficient $R$ and relative error AARE of test group calculated according to Equations (12) and (13). It can be found that the polynomial fitting with order 4 is the worst, the correlation coefficient $R$ is only 0.9121, and the AARE is as high as 18.3%. The prediction accuracy of order 3,5,6 is better, the best one is order 5, the correlation coefficient $R$ is 0.9895, and the average relative error is 8.048%.

$$R = \frac{\sum_{i=1}^{N}\left(E_i - \overline{E}\right)\left(P_i - \overline{P}\right)}{\sqrt{\sum_{i=1}^{N}\left(E_i - \overline{E}\right)^2}\sqrt{\sum_{i=1}^{N}\left(P_i - \overline{P}\right)^2}} \tag{12}$$

$$\mathrm{AARE}(\%) = \frac{1}{N}\sum_{i=1}^{N}\left|\frac{E_i - P_i}{E_i}\right| \times 100\% \tag{13}$$

**Table 3.** Correlation coefficient and relative average error corresponding to different orders.

| Order | *R* | *AARE* (%) |
|-------|------|------------|
| 3 | 0.9884 | 8.184 |
| 4 | 0.9121 | 18.298 |
| 5 | 0.9895 | 8.048 |
| 6 | 0.9894 | 8.052 |

Figure 7 is a scatter diagram composed of experimental stress and calculated stress. The scatter diagram can more intuitively show the prediction ability of the strain correction constitutive equation. The closer the scatter point is to the straight line y = x, the polynomial fitting of order 4 can significantly deviate from the prediction of stress scatter point y = x, which further proves that the previous evaluation of order 4 has the largest relative error.

Figure 8 shows the case of fitting four material constants ($\alpha$, $n$, $Q_{act}$, $\ln A$) with polynomial of order 5, replacing $Z$ parameter with the form containing $Q_{act}$ in Equation (5). The latter half of Equation (14) is an explanation of the four material constants in the first half of Equation (14). Where the coefficients of the fifth polynomials in Equation (14) with respect to the four material constants are shown in Table 4.

$$\sigma = \frac{1}{\alpha(\varepsilon)} \ln \left\{ \left( \frac{\dot{\varepsilon} \exp(\frac{Q(\varepsilon)}{RT})}{A(\varepsilon)} \right)^{\frac{1}{n(\varepsilon)}} + \left[ \left( \frac{\dot{\varepsilon} \exp(\frac{Q(\varepsilon)}{RT})}{A(\varepsilon)} \right)^{\frac{2}{n(\varepsilon)}} + 1 \right]^{\frac{1}{2}} \right\}$$

$$\begin{cases} \alpha(\varepsilon) = B_0 + B_1\varepsilon + B_2\varepsilon^2 + B_3\varepsilon^3 + B_4\varepsilon^4 + B_5\varepsilon^5 \\ n(\varepsilon) = C_0 + C_1\varepsilon + C_2\varepsilon^2 + C_3\varepsilon^3 + C_4\varepsilon^4 + C_5\varepsilon^5 \\ \ln A(\varepsilon) = D_0 + D_1\varepsilon + D_2\varepsilon^2 + D_3\varepsilon^3 + D_4\varepsilon^4 + D_5\varepsilon^5 \\ \frac{Q(\varepsilon)}{1000} = F_0 + F_1\varepsilon + F_2\varepsilon^2 + F_3\varepsilon^3 + F_4\varepsilon^4 + F_5\varepsilon^5 \end{cases} \tag{14}$$

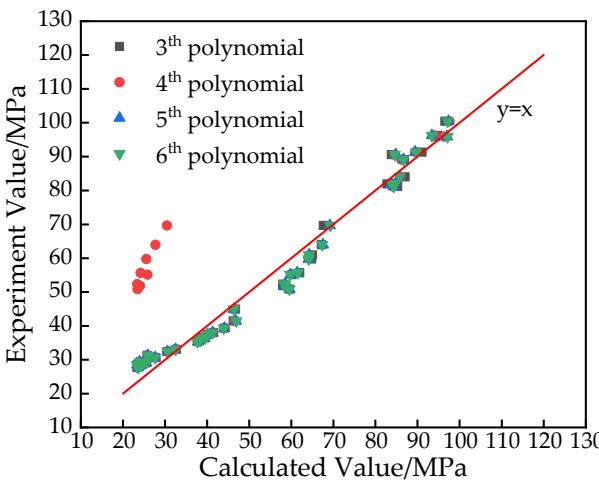

**Figure 7.** Experimental stress and calculated stress of different orders high quality 20MnCr5(SH) gear steel.

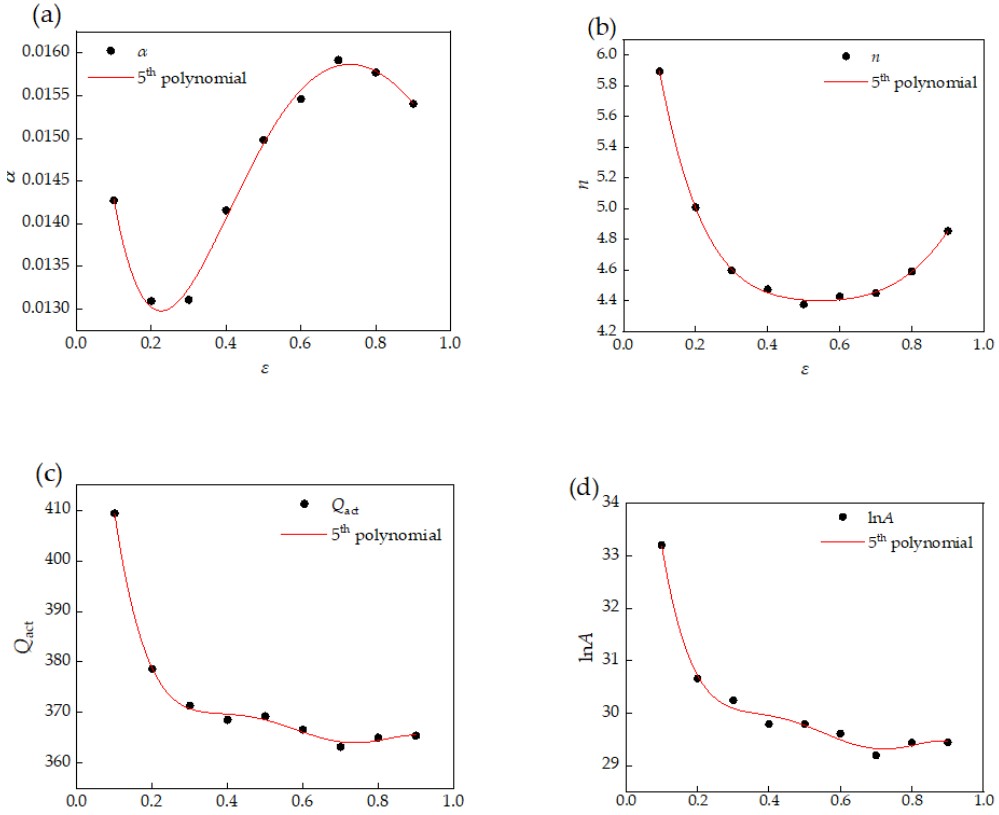

**Figure 8.** Constitutive constants at different strains. (**a**) $\alpha$; (**b**) $n$; (**c**) $Q$; (**d**) $\ln A$.

**Table 4.** Coefficients of polynomial fitting curves for material coefficients.

|   | 0 | 1 | 2 | 3 | 4 | 5 |
|---|---|---|---|---|---|---|
| B | 0.0183 | −0.0599 | 0.2261 | −0.3413 | 0.2348 | −0.0633 |
| C | 7.6297 | −23.2061 | 67.5240 | −98.6714 | 70.6542 | −18.6440 |
| D | 39.9165 | −99.8983 | 395.4466 | −757.2713 | 688.5364 | −238.0469 |
| F | 490.3597 | −1187.4711 | 4549.0954 | −8382.6452 | 7348.1034 | −2457.3164 |

## 5. Processing Maps

Processing maps are effective tools to characterize the hot working performance of materials. Based on the superposition of processing efficiency map and plastic instability map, the processing efficiency is expressed by contour lines, and the plastic instability area is divided. The theory of hot working graph is based on the dynamic material model (DMM) proposed by Prasad et al. [27]. The processing efficiency graph itself is a ternary function ($\eta$ with respect to $\sigma$, $\dot{\varepsilon}$, $\varepsilon$), so a complete processing efficiency graph cannot be drawn. Therefore, the processing efficiency graph reflects the power dissipation under a certain strain; that is, the ternary function graph with respect to stress and strain rate. DMM theory assumes that the power absorbed by metal deformation is consumed by metal deformation and microstructure evolution, respectively. The power consumed by plastic deformation is recorded as $G$, and the power consumed by microstructure evolution is recorded as J. the relationship between the power absorbed by thermal deformation and $G$, $J$, and the calculation method of $G$, $J$ are shown in Equation (15)

$$P = \dot{\varepsilon} \cdot \sigma = G + J = \int_0^{\varepsilon} \sigma d\dot{\varepsilon} + \int_0^{\sigma} \varepsilon d\sigma \tag{15}$$

DMM theory points out that the relationship between work hardening coefficient $m$ and flow stress $\sigma$, strain rate $\dot{\varepsilon}$ is as follows [28,29]:

$$m = \frac{\partial(\ln \sigma)}{\partial(\ln \dot{\varepsilon})} \tag{16}$$

In the process of hot compression deformation, the relationship between work efficiency $\eta$ and work hardening coefficient $m$ is shown in Equation (17) [30,31]:

$$\eta = \frac{J}{J_{\max}} = \frac{2m}{m+1} \tag{17}$$

Not only do dynamic recovery and dynamic recrystallization occur, but also possible plastic damage (such as adiabatic shear band, plastic cracking, etc.) in processing efficiency $\eta$. Therefore, Kumar et al. [32,33] proposed the instability criterion based on DMM theory:

$$\zeta(\dot{\varepsilon}) = \frac{\partial \ln\left(\frac{m}{m+1}\right)}{\partial \ln \dot{\varepsilon}} + m < 0 \tag{18}$$

According to the instability criterion, when $\zeta(\dot{\varepsilon}) < 0$, this region is the instability region.

The calculation results of strain rate sensitivity $m$ under various deformation conditions with strain variables of 0.1, 0.2, 0.3, 0.4, 0.5 and 0.6 are shown in Table 5.

According to the data in Table 5, the power dissipation rate under various deformation conditions can be calculated and the three-dimensional diagram of power dissipation rate can be drawn. As shown in Figure 9, the selected strains are 0.1, 0.2, 0.3, 0.4, 0.5 and 0.6, respectively, which can reflect the change trend of power dissipation rate during thermal deformation. The higher the power dissipation rate under different deformation conditions, the higher the power absorbed by recrystallization and microstructure evolution, and the more suitable for thermoplastic processing. In Figure 9a, the maximum power dissipation rate is 0.29, and there are two peaks of power dissipation rate, one in the region centered at 1050 °C and strain rate of 0.13 s$^{-1}$, and the other in the triangular region centered at

1230 °C and strain rate of 0.002 s$^{-1}$. In Figure 9b, the peak region of power dissipation rate is gradually connected, and the region is approximately between 1050 °C and 1230 °C, the strain rate is between 0.13 s$^{-1}$ and 0.37 s$^{-1}$ and the maximum value of power dissipation rate is 0.33. In Figure 9c–f, the maximum value of power dissipation rate is above 0.38. The peak area ranges from 1030 °C to 1230 °C, and the strain rate ranges from 0.37 s$^{-1}$ to 2 s$^{-1}$. In Figure 9a–f, there is a same Valley region, which is located in the temperature range of 930–1130 °C and the strain rate range of 0.002–0.02 s$^{-1}$. When the strain is 0.1, the power dissipation rate is as low as 0.01, which is the smallest of all the troughs in Figure 9a–f.

**Table 5.** The value of *m* of each computing node at different deformation conditions.

| $\varepsilon$ | $\dot{\varepsilon}/s^{-1}$ | $T/°C$ | | | |
|---|---|---|---|---|---|
| | | 930 | 1030 | 1130 | 1230 |
| 0.1 | 0.002 | 0.0997 | 0.0010 | 0.1327 | 0.1715 |
| | 0.02 | 0.1015 | 0.1086 | 0.1325 | 0.1564 |
| | 0.2 | 0.0827 | 0.1653 | 0.1358 | 0.1428 |
| | 2 | 0.0621 | 0.1145 | 0.1394 | 0.1443 |
| 0.2 | 0.002 | 0.1199 | 0.0218 | 0.1071 | 0.1267 |
| | 0.02 | 0.1247 | 0.1432 | 0.1662 | 0.1579 |
| | 0.2 | 0.0948 | 0.1884 | 0.1821 | 0.1925 |
| | 2 | 0.0601 | 01120 | 0.1388 | 0.1959 |
| 0.3 | 0.002 | 0.1438 | 0.0144 | 0.1085 | 0.1231 |
| | 0.02 | 0.1423 | 0.1361 | 0.1599 | 0.1292 |
| | 0.2 | 0.1025 | 0.2124 | 0.2036 | 0.2073 |
| | 2 | 0.0641 | 0.1671 | 0.1958 | 0.2793 |
| 0.4 | 0.002 | 0.1436 | 0.0142 | 0.1155 | 01275 |
| | 0.02 | 0.1472 | 0.1268 | 0.1559 | 0.1411 |
| | 0.2 | 0.1161 | 0.2186 | 0.2017 | 0.1945 |
| | 2 | 0.0813 | 0.1978 | 0.2070 | 0.2342 |
| 0.5 | 0.002 | 0.1449 | 0.0064 | 0.1099 | 0.1111 |
| | 0.02 | 0.1551 | 0.1172 | 0.1468 | 0.1374 |
| | 0.2 | 0.1265 | 0.2205 | 0.1994 | 0.1958 |
| | 2 | 0.0877 | 0.2130 | 0.2150 | 0.2279 |
| 0.6 | 0.002 | 0.1447 | 0.0045 | 0.1109 | 0.1044 |
| | 0.02 | 0.1475 | 0.1086 | 0.1321 | 0.1341 |
| | 0.2 | 0.1277 | 0.2202 | 0.1985 | 0.1940 |
| | 2 | 0.1050 | 0.2276 | 0.2437 | 0.2242 |

The processing maps of 20MnCr5(SH) are given in Figure 10, when the strain rate is 0.002–2 s$^{-1}$ and 930–1230 °C, and the strain $\varepsilon$ is 0.1, 0.2, 0.3, 0.4, 0.5 and 0.6, respectively. By superposition of power dissipation diagram and instability criterion, the processing map is obtained. The gray area on the map is an unstable area. As shown in Figure 10a, when $\varepsilon$ = 0.1, there are two unstable regions, one located in the "triangle area" of 930–1050 °C and 0.13–2 s$^{-1}$, the other located at 1150–1200 °C, and the strain rate is 0.002–0.018 s$^{-1}$. In Figure 10b, when $\varepsilon$ = 0.2, the instability region is concentrated in the "triangle region" with a strain rate of 0.13–2 s$^{-1}$ and 930–1100 °C. The area of the instability region in Figure 10b to e decreases with the increase of the stress variable, and the instability region disappears in Figure 10f. According to the trend of instability region in Figure 10, it is shown that with the increase of the strain, the area of the instability area is smaller and smaller, and finally disappears. According to the power dissipation maps and the instability criterion analysis, the temperature range of strain rate is 0.05 s$^{-1}$–1 s$^{-1}$, 1030–1100 °C, and it is in the peak area of power diffusion rate, which is conducive to obtaining good microstructure of thermal processing.

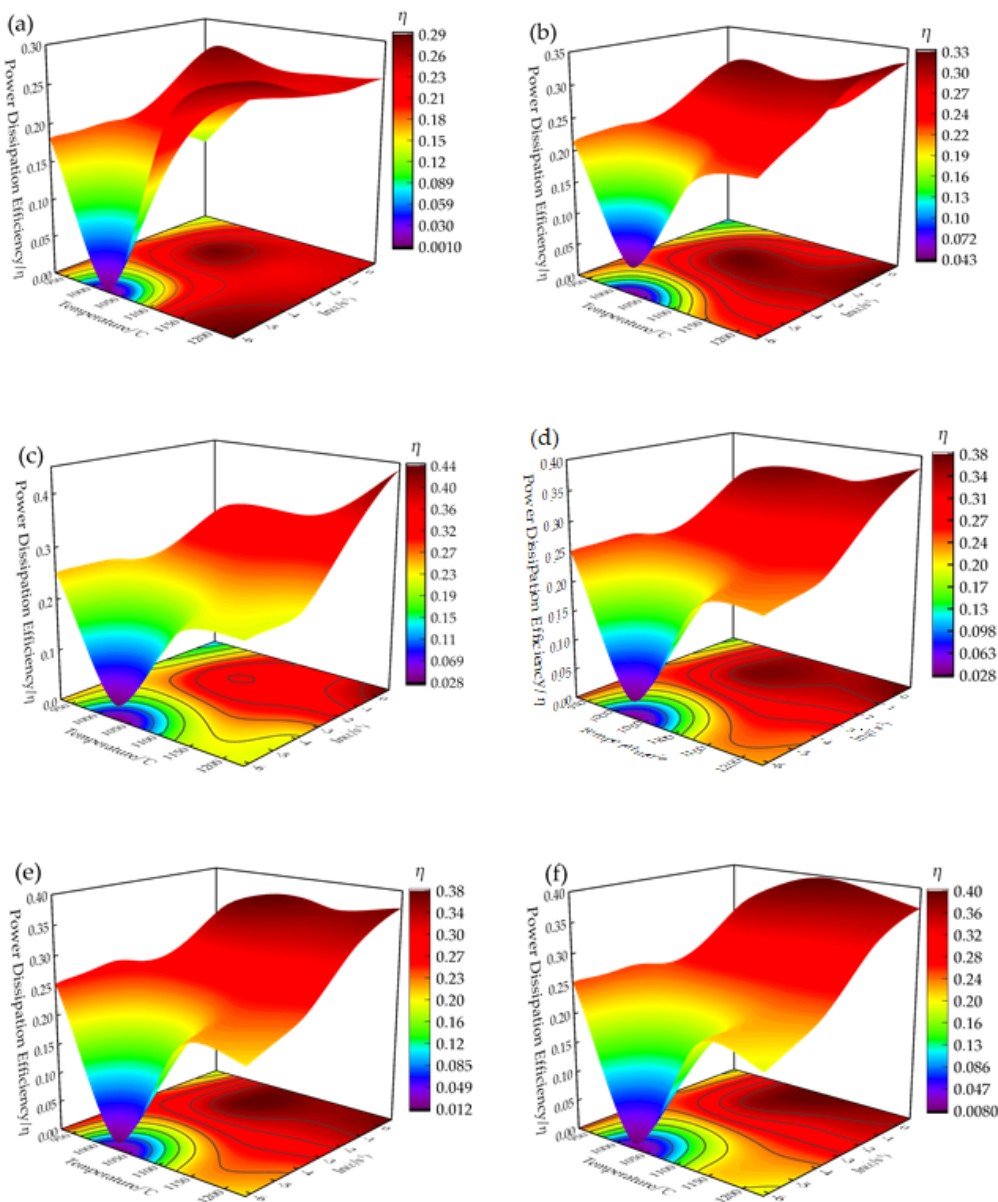

**Figure 9.** Power dissipation maps for the 20MnCr5(SH) gear steel at strains of (**a**) 0.1; (**b**) 0.2; (**c**) 0.3; (**d**) 0.4; (**e**) 0.5; (**f**) 0.6.

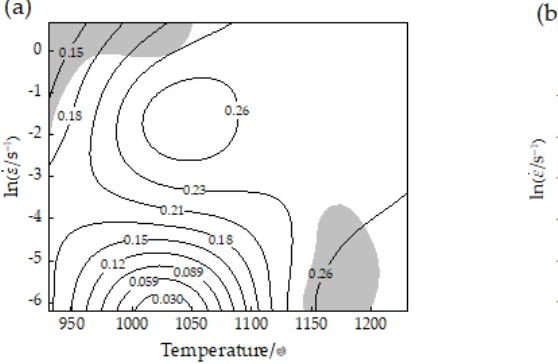

**Figure 10.** *Cont.*

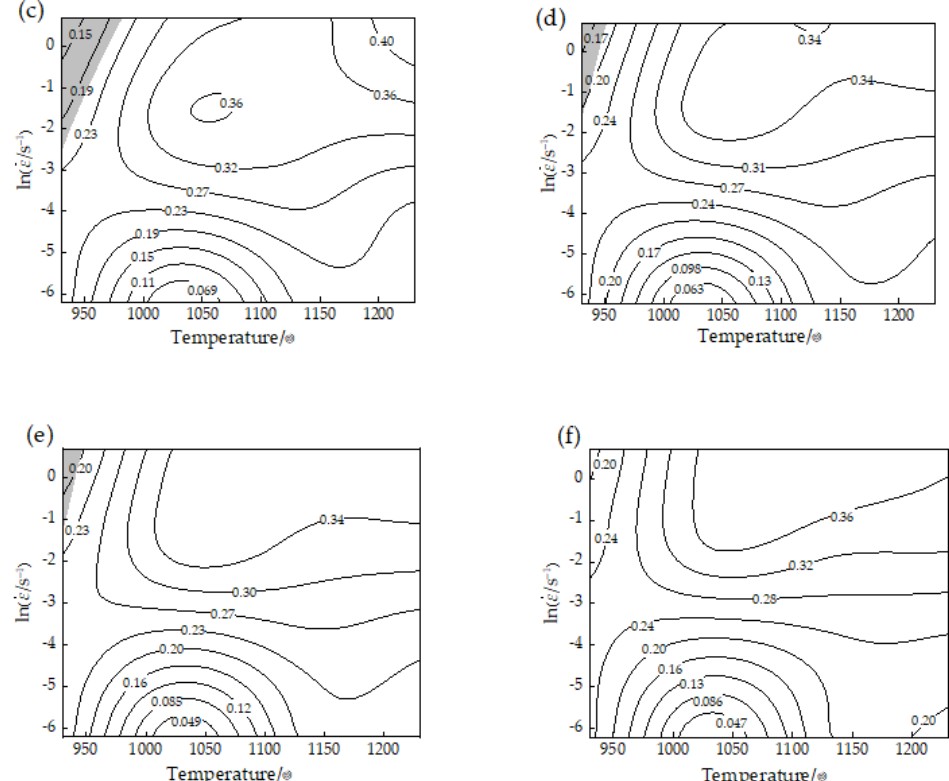

**Figure 10.** Processing maps for the 20MnCr5(SH) gear steel at strains of (**a**) 0.1; (**b**) 0.2; (**c**) 0.3; (**d**) 0.4; (**e**) 0.5; (**f**) 0.6.

To summarize, when processing high quality 20MnCr5(SH) gear steel by hot plasticity, it should refer to the processing maps, avoid the plastic instability zone and try to select the area with high processing efficiency zone, so as to obtain good forging structure and improve the comprehensive mechanical properties and quality of the product.

## 6. Conclusions

(1)　The higher the hot deformation temperature of 20MnCr5(SH) gear steel is, the smaller the flow stress is, and vice versa; the lower the strain rate is, the smaller the flow stress is, and vice versa. The observation of metallographic structure shows that dynamic recrystallization occurs during hot deformation of 20MnCr5(SH) gear steel. The austenite grains are filled with martensite lath bundles.

(2)　According to the true stress-strain curve, the parameters of the constitutive equation are obtained by linear fitting method, and the 20MnCr5(SH) constitutive equation is obtained. The thermal deformation activation energy $Q_{act}$ of 20MnCr5(SH) is 356.412 kJ/mol, and the strain rate sensitivity index $n$ is 4.802. Dynamic recovery and dynamic recrystallization are the main softening mechanisms of 20MnCr5(SH).

(3)　The strain modified constitutive equation of 20MnCr5(SH) is established. The material constants ($\alpha$, $n$, $Q_{act}$, $\ln A$) involved in the constitutive equation are fitted by five polynomial. The flow stress calculated by strain modified constitutive equation is compared with the measured value of thermal pressure test. The correlation $R = 0.9895$, and the average relative error is 8.048%. It shows that the strain modified constitutive equation has strong stress prediction ability.

(4)　Based on the dynamic material model and the instability criterion, the processing maps of strain $\varepsilon = 0.1$, 0.2, 0.3, 0.4, 0.5 and 0.6 are established. The results show that the best processing parameters of 20MnCr5(SH) gear steel are strain rate 0.05 s$^{-1}$–1 s$^{-1}$, temperature 1030–1100 °C. In this region, the plastic instability region can be avoided and the power dissipation rate is large.

**Author Contributions:** J.Y. is responsible for designing experiments and writing manuscripts, L.W. is responsible for funding and revising manuscripts, Y.Z. is responsible for metallographic experiment and analysis, and Z.Z. is responsible for material hot compression experiment and data analysis. All authors have read and agreed to the published version of the manuscript.

**Funding:** This research was funded by [National Natural Science Foundation of China] grant number [51865057].

**Institutional Review Board Statement:** Not applicable.

**Informed Consent Statement:** Not applicable.

**Data Availability Statement:** The data that support the findings of this study are available from the corresponding author upon reasonable request.

**Conflicts of Interest:** The authors declare no conflict of interest.

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
