# Peer review of "Strain Modified Constitutive Equation and Processing Maps of High Quality 20MnCr5(SH) Gear Steel"

_crystals, doi:10.3390/cryst11050536_

Round 1

Reviewer 1 Report

After the corrections, the article may be in the scope of interest to the readers of the Crystals journal.

Author Response

       Thank you for your comments on our paper! We found the activation energy calculation error through self inspection, and recalculated and corrected the physical quantities related to activation energy. You are welcome to continue to put forward your valuable opinions. Thank you again!

Reviewer 2 Report

The manuscript added the parameters of different strain into the constitutive equations based on the experimental data.

There are quite a lot of problems in writing, but at the current stage, I mainly want to point out the constitutive constants at different strains, Figure 7. One has to consider whether such polynomial fitting up to 6 orders really makes any sense. The higher-order numerical fitting normally has the problem of so-called over-fitting. That means, when one uses the fitted formula on new data, the fitted formula would have very poor behaviour.  So Figure 8 does not really make sense since the modified constitutive equation is modified based on the experimental data the authors already have, so how bad it could be. One should check whether the modified constitutive equation based on new experimental data.   

Author Response

Dear reviewer:

I read your opinion carefully, mainly about the unreliability of high-order fitting, and the verification can not be verified by the data of this experiment.

I think what you said is really reasonable. The two problems you raised are the problems that will be encountered in the process of establishing the constitutive equation of metal materials. After reading a lot of literature in this field, I found that five times fitting of four material constants is quite common in today's strain correction constitutive equation of metal materials, so I changed six times fitting to five times fitting, so as to solve your problem of over fitting.

Also after reading a lot of literature, in the verification of constitutive equation, the authors have used the original data to verify the correctness of the constitutive model. I think the verification error of constitutive equation here is to test the error caused by multiple fitting in the process of establishing the mathematical model. From this point of view, I think figure 8 has certain significance. Of course, there is some truth in what you said, but this method is adopted in the current literature on the constitutive equation of metal materials.

Thank you very much for your suggestions on this paper. If you have any suggestions, please continue to put forward and thank you again!

The literature involved in the above process is as follows:

Doi:10.3390/met10060782

Doi:10.3390/met10060828

https://doi.org/10.3390/met11010075

https://doi.org/10.3390/met11010077

Reviewer 3 Report

In the manuscript, the authors investigated the high-temperature deformation behavior of 20MnCr5(SH) gear steel. The main results are the proposed strain modified Arrhenius constitutive equation and processing maps. The optimal mode of steel processing was proposed based on the processing map analysis. The manuscript is well structured, the conclusions are substantiated. 
I have a few comments:
1.    Line 104. Instead of the phrase «Gleeble-1500D thermal simulator» there should be a «Gleeble 1500D thermo-mechanical simulator»
2.    Line 109. Instead of the phrase « strain rate was 0.916.» there should be a « strain was 0.916»
3.    In the experimental procedure, the authors must indicate the brand of the grinding and polishing machine and the composition of the etchant.
4.    Figure 4. The grain boundaries are not visible in most of the figure. Authors should use a high-quality image of the microstructure. 

Author Response

I have to say that you have been very careful in the process of reviewing our paper and observed a lot of details. I will revise our paper as follows:

1. Line 104. Instead of the phrase « Gleeble-1500D thermal simulator » there should be a « Gleeble 1500D thermo-mechanical simulator » Reply: this question has been changed.

2. Line 109. Instead of the phrase « strain rate was 0.916. » there should be a « strain was 0.916 » Reply: this writing error has also been changed.

3. In the experimental procedure, the authors must indicate the brand of the grinding and polishing machine and the composition of the etchant.

Reply: the name of the equipment you requested, the composition and preparation method of the corrosive agent have been added.

4. Figure 4. The grain boundaries are not visible in most of the figure. Authors should use a high-quality image of the microstructure.

Reply: the grain boundary in the gold phase diagram is not clear, and the metallographic picture with clearer grain boundary has been replaced

Thank you again!!

Round 2

Reviewer 2 Report

I am happy to see that the authors agree with the problem I pointed out. I appreciate the publications the authors found. I am aware that there are published papers doing a similar fitting. This actually makes me feel quite bad in the sense that we are aware of the problem, but we keep doing the same thing (or we keep making the same mistake) simply because other people did that before and their paper got published.... Following this logic, if the first one who did this was wrong, we will have to be wrong.  I do not want to reject the paper simply because I personally think the fitting is inappropriate while other papers doing the same thing got published somewhere else, but this problem really bothers me......So I strongly suggest the authors carry out the study of fitting sensitivity: As we agree that, higher-order fittings look good for the data used for fitting, but may give a bad prediction on the new data.  A good/proper numerical fitting formulas should be the compromise between the fitting accuracy (evaluated based on the data used for fitting) and data predictability (and in fact, the predictability is something we really care about). So the authors should follow a similar data processing technique commonly used in the machine learning community: split the whole data (all data you used for fitting in the manuscript) as the training data and the test data. For example, a 80/20 split, means: use 80% of the data you have to fit the parameter and use 20% of the data to verify the accuracy of the fitted function/parameter. Of course one can try different orders of polyfit, but the ultimate goal is to find out the fittings that give the best prediction (minimum error) on the test data.  By doing this, we will not be bothered anymore about the orders and accuracy of the fitting, because essentially you have done the optimization of the fitting.

Author Response

Dear reviewer

I've read your suggestion carefully. You're right. We can't imprison ourselves because of other people's imperfections. After all, the truth is in the hands of a few people.

According to your suggestion, I use 25% of the data to test the prediction accuracy of the strain correction constitutive equation, and 75% of the data to build the strain correction constitutive equation. The selected test group was all stress data at 1130 ℃. The research scope of polynomial order is 3,4,5,6, because order less than 3 obviously can not meet the needs of fitting accuracy, order higher than 6 may cause over fitting. Taking correlation coefficient and average relative error as two parameters to evaluate the prediction accuracy, it is found that the fitting effect of order 4 is the worst, and that of order 5 is the best. We decided to use the polynomial of order 5 to express the relationship between the four material constants and strain.

Thank you again for your suggestion. Your suggestion not only makes the article more perfect, but also makes the proposed modification scheme very practical. We will continue to use this verification idea in the future article writing.

Round 3

Reviewer 2 Report

I appreciate the changes the authors made, I have no more suggestions on the manuscript.